You can’t teach speed: sprinters falsify the deliberate practice model of expertise

Lombardo Michael P. 1 lombardm@gvsu.edu
Deaner Robert O. 2
1 Department of Biology, Grand Valley State University , Allendale, MI , USA
2 Department of Psychology, Grand Valley State University , Allendale, MI , USA
Mueller Shane
Electronic publication date: 2014 Jun 26
Publication date: 2014
Volume: 2
Electronic Location ID: e445
Received 2014 Apr 11; Accepted 2014 Jun 2
Copyright: © 2014 Lombardo and Deaner
Copyright year: 2014
Copyright holder: Lombardo and Deaner
License: This is an open access article distributed under the terms of the Creative Commons Attribution License, which permits unrestricted use, distribution, reproduction and adaptation in any medium and for any purpose provided that it is properly attributed. For attribution, the original author(s), title, publication source (PeerJ) and either DOI or URL of the article must be cited.
License URL: https://creativecommons.org/licenses/by/4.0/

Keywords: Expertise, Deliberate practice model of expertise, Athletic performance, Sprinting, Evolutionary psychology, Display, Talent, Running, Sports, Training

Funding: This project was supported by the Departments of Biology and Psychology at Grand Valley State University. The funders had no role in study design, data collection and analysis, decision to publish, or preparation of the manuscript.

==============================
Many scientists agree that expertise requires both innate talent and proper training. Nevertheless, the highly influential deliberate practice model (DPM) of expertise holds that talent does not exist or makes a negligible contribution to performance. It predicts that initial performance will be unrelated to achieving expertise and that 10 years of deliberate practice is necessary. We tested these predictions in the domain of sprinting. In Studies 1 and 2 we reviewed biographies of 15 Olympic champions and the 20 fastest American men in U.S. history. In all documented cases, sprinters were exceptional prior to initiating training, and most reached world class status rapidly (Study 1 median = 3 years; Study 2 = 7.5). In Study 3 we surveyed U.S. national collegiate championships qualifiers in sprinters (n = 20) and throwers (n = 44). Sprinters recalled being faster as youths than did throwers, whereas throwers recalled greater strength and throwing ability. Sprinters’ best performances in their first season of high school, generally the onset of formal training, were consistently faster than 95–99% of their peers. Collectively, these results falsify the DPM for sprinting. Because speed is foundational for many sports, they challenge the DPM generally.

“I can make you faster, but I can’t make you fast.”

Jerry Baltes, Head Coach, Grand Valley State University cross-country and track and field

Introduction

A major scientific goal is identifying the factors that contribute to expertise or superior performance in domains such as dancing, decision-based games (e.g., chess), music, professional decision-making (e.g., medical diagnosis), and sports.

Many contemporary scientists hold that the phenotypic expression of traits, including those associated with expertise, reflect an interaction between genes (i.e., nature) and experience (i.e., nurture) (Pinker, 2002; Ridley, 2003). However, some hold that experience might, by itself, be sufficient to account for expertise (Ericsson, Krampe & Tesch-Römer, 1993; Ericsson, Prietula & Cokely, 2007; Ericsson, Nandagopal & Roring, 2009; Howe, Davidson & Sloboda, 1998). These scholars have developed and endorsed a deliberate practice model (DPM) of expertise which emphasizes the crucial role of deliberate practice, defined as training that is structured, attentive, maximally effortful, and subject to immediate feedback from a coach (Ericsson, Krampe & Tesch-Römer, 1993; Ericsson, Prietula & Cokely, 2007; Ericsson, Nandagopal & Roring, 2009; Howe, Davidson & Sloboda, 1998). The DPM, at least the strong version addressed here, holds that consistent long period of deliberate practice (about 10 years or 10,000 h) is necessary and sufficient for any healthy individual to achieve expert performance in any domain (Ericsson, Krampe & Tesch-Römer, 1993; Ericsson, Prietula & Cokely, 2007; Ericsson, Nandagopal & Roring, 2009). The only acknowledged exceptions are for domains where success is dependent on body size and/or height, such as basketball (Visscher, 2008; Livshits et al., 2002). Representative claims by DPM advocates are provided in Table S1.

The DPM has been enormously successful in stimulating research on the mechanisms (e.g., physiological, motor, perceptual, cognitive) and the kinds of training (e.g., social or solitary, leisurely or concentrated) that underlie expert performance (Baker, Côte & Abernathy, 2003; Gagné, 2009; Starkes et al., 1996; Williams & Ford, 2008). This impact is revealed by the fact that as of 10 April 2014, the foundational DPM paper by Ericsson, Krampe & Tesch-Römer (1993) has been cited over 4200 times on Google Scholar. The DPM’s influence extends beyond academia, as it has served as the basis for several popular trade books (Coyle, 2009; Gladwell, 2008; Syed, 2010).

Despite the DPM’s popularity, there are strong theoretical and empirical reasons to doubt its fundamental claim that deliberate practice is necessary and sufficient for achieving expertise. First, from a theoretical perspective, the assumptions of the DPM are inconsistent with mounting literature documenting the influence of an individual’s genotype on its behavior (Plomin et al., 2008; Ebstein et al., 2010) and physiological performance (Collins, 2009; Timmons et al., 2010; Hagberg et al., 2011). In addition, the DPM may not adequately explain or predict the development of expert performance in motor dominated domains because of its emphasis on cognitive mechanisms (Abernethy, Farrow & Berry, 2003). Moreover, evolutionary hypotheses for the existence of display or “show-off” expertise domains hold that, in large part, expert performances are impressive precisely because they function to signal heritable genetic variation (Deaner, 2013; de Block & Dewitte, 2009; Lombardo, 2012; Miller, 2000; Hawkes & Bird, 2002).

Second, empirical work indicates that foundational abilities for some expertise domains have a heritable genetic basis (Hambrick & Meinz, 2011; Tucker & Collins, 2012). For instance, working memory capacity is highly heritable (Kremen et al., 2007) and, even when the amount of deliberate practice is controlled, it predicts performance in poker (Meinz et al., 2012) and piano playing (Meinz & Hambrick, 2010). Similarly, maximal oxygen consumption (VO2max) is crucial for elite performance in endurance sports, and both untrained VO2max and VO2max responsiveness are highly heritable (Bouchard et al., 2011; Bouchard et al., 2012).

Third, scientists have noted weaknesses in the behavioral evidence that directly addresses the DPM’s claims. The DPM is based on correlational studies showing that achievement is strongly correlated with accumulated deliberate practice. One problem with the DPM is that it assumes that deliberate practice drives the correlation, yet it is possible that innate ability, or talent, is causal (Ackerman, 2013). In other words, individuals that experience early success as a result of superior innate ability typically become more motivated to train (Howe, Davidson & Sloboda, 1998). For example, in the domain of music expertise, Ruthsatz et al. (2008) reanalyzed the data in Ericsson, Krampe & Tesch-Römer (1993) and showed that, even as young children, the violinists who would eventually accumulate a large amount of deliberate practice (about 10,000 h on average) and become elite were already more likely than others to win competitions despite training for similar durations as those who would not become not become as accomplished.

A fourth problem with the DPM is its claim that deliberate practice explains a very high proportion of the variance in the attainment of expertise; the empirical data contradict this (Hambrick et al., 2013). For example, deliberate practice explained only 28% of performance variation among dart players (Duffey, Baluch & Ericsson, 2004). Among chess players, deliberate practice explained only 34% of performance variation. And in fact, some chess players did not reach the master level despite 25,000 h of practice, whereas others reached this level with only 3,000 h of practice (Gobet & Campitelli, 2007). Similarly, a study of 459 elite Australian athletes from 34 different sports demonstrated that the mean period of development from novice to elite athlete was 7.5 ± 4.1 (SD) years, and 69% of athletes in individual sports achieved elite status in less than five years (Oldenziel, Gagne & Gulbin, 2004).

Although these appear to be intractable empirical problems for the DPM’s strong claims, DPM proponents have presented counter-arguments (Ericsson, Krampe & Tesch-Römer, 1993; Ericsson, Nandagopal & Roring, 2009; Howe, Davidson & Sloboda, 1998; Ericsson, 2013). For example, Ericsson, Nandagopal & Roring (2009) and Ericsson (2013) disputed the heritability of VO2max and other physiological factors relevant to sports performance (Collins, 2009; Timmons et al., 2010; Tucker & Collins, 2012; Bouchard et al., 2011; Bouchard et al., 2012). In addition, Howe, Davidson & Sloboda (1998) noted that modest correlations between deliberate practice and achievement might reflect unequal quality of instruction or effectiveness of practice strategy. In sum, then, there remains uncertainty regarding the validity of the DPM.

Here we provide strong tests of two critical DPM predictions in the domain of sprinting (e.g., footraces over short distances such as 100 m). First, we tested the DPM’s prediction that initial performance in a domain (i.e., prior to deliberate practice) and final performance in the domain will be unrelated. Although there has been much discussion about prodigies, since their occurrence would falsify the DPM, it is impossible to assess whether an individual is exceptional prior to training in most domains (Ericsson, Krampe & Tesch-Römer, 1993; Howe, Davidson & Sloboda, 1998). For example, it makes little sense to ask, much less measure, how gifted a child is at playing chess before they have become knowledgeable about the rules of the game. In the domain of sprinting, however, it is possible to assess performance prior to training. This is because nearly all children run in the course of normal play. Thus, a child who is an exceptionally fast runner can readily assess their ability relative to their peers, as can adult observers. The DPM implies that initial performance in a domain represents random error and that only formal training determines an individual’s ultimate level of performance. In contrast, an interactive “talent matters” framework predicts that, as children, most elite sprinters will have been fast relative to their peers and that these individuals will have performed exceptionally well as soon as they began formal competition and training.

Second, we tested the DPM’s prediction that roughly 10 years of deliberate practice are required to reach expert status. Although some studies indicate that some athletes reach expert status with substantially less than 10 years of training (Oldenziel, Gagne & Gulbin, 2004), several others are consistent with the “10 year rule” (Starkes et al., 1996; Kalinowski, 1985; Wallingford, 1975; Monsaas, 1985; Helsen, Starkes & Hodges, 1998; Hodges & Starkes, 1996; Young & Salmela, 2002). In addition, a limitation of most studies is that there is some degree of subjectivity in the rating of expertise. For example, factors related to team selection (e.g., a coach preferring older players (Cobley et al., 2009)) may mean that a soccer or field hockey player may not play for his or her national team (and thus be classified as an expert) until his or her mid-20 s despite having the ability to do so several years earlier. Because sprinting expertise is based on objective performances, it provides an unusually strong test of the DPM’s main prediction that 10 years or 10,000 h of deliberate practice are required to achieve expert performance.

We tested the two key predictions of the DPM with three complementary studies. In Study 1 we reviewed the biographies of male and female Olympic sprint champions. In Study 2 we reviewed the biographies of the 20 fastest male 100 m runners in U.S. history. In Study 3 we surveyed male and female sprinters who qualified for the 2012 U.S. collegiate national championships. To our knowledge, these are the first studies to address the DPM in sprinting.

Study 1: Biographies of Olympic champions

We examined the biographies of Olympic champions because becoming an Olympic champion shows unambiguous evidence of expertise. Moreover, because there is often great interest in sprint champions, biographies have been written about many of them. These generally include detailed information on the sprinter’s athletic development, making them ideal for addressing the predictions of interest.

Although our main focus was testing the two predictions discussed above, we also explored whether champion sprinters had participated in organized sports prior to beginning their sprinting career. This was important because it could be argued that exceptional sprinting performance prior to formal sprint training reflects skill transfer from other sports (Baker, Côte & Abernathy, 2003; Smeeton, Ward & Williams, 2004).

Methods

We sought English-language biographies, including autobiographies, published in print of male and female gold medalists in the 100 m or 200 m sprints from the 1896 to 2012 Olympic Games. We were able to obtain at least one biography for 15 sprinters (8 women) and obtained two or more biographies for six sprinters. Two of the champion sprinters, Ben Johnson in 1988 and Marion Jones in 2000, were later stripped of their gold medals due to their use of performance enhancing drugs (PEDs). We retained these sprinters in the sample because the available information indicated that they reached world class status before they began using PEDs. Furthermore, the use of PEDs may be common among world class sprinters, even those who are never sanctioned (Francis & Coplon, 1991; Moore, 2012).

From the biographies, we recorded any evaluation of the sprinter being exceptional or unexceptional relative to their peers. We recorded who made the evaluation, the sprinter, a teacher, or a coach, or another individual. We recorded the sprinter’s age when the evaluation occurred and the age when they began formal training with a coach. We assumed that formal training with a coach would indicate the onset of training activities that would best correspond with “deliberate practice.” In some cases, the sprinter’s age at the time of first evaluation or first formal training was not mentioned, but their grade in school was, and this allowed us to estimate their age. For instance, the first year of high school was assumed to indicate being age 14 years. In cases where there was no explicit mention of the initiation of formal training, we assumed this occurred at the onset of formal competition, usually in the first year of high school. We also noted any mention of a sprinter’s formal participation, or not, in an organized sport other than track and field prior to beginning formal sprint training. We also recorded the sprinter’s age when they first represented their country in the senior (i.e., open to all ages) World Championships or Olympic Games. We considered national representation indicative of achieving world class or expert status. These are highly selective, conservative measures of expertise because these championships do not occur every year and individuals who have reached world class performance levels may not qualify for them due to injury or other issues.

Results and discussion

All 15 Olympic champion sprinters were recognized as being exceptionally fast relative to their peers before or coincident with their initiation of formal training. There was no indication in any biography that any sprinter was initially unexceptional. We condensed key information in Table 1 and summarized relevant passages from each biography in Table S2.

Table 1 Male and female 100 m and 200 m Olympic gold medal winners for which autobiographies or biographies published in print were available.

Athlete	Sex	Olympic
games	Events won	Superior sprinting
speed documented
as youth or
teenager	Years of DP
to reach
world class
statusa	Reference	
Jesse Owens	M	1936	100 m, 200 m	Yes	4	Baker (1986)	
Helen Stephens	F	1936	100 m	Yes	3	Hanson (2004)	
Wilma Rudolph	F	1960	100 m, 200 m	Yes	2	Smith (2006), Mallon (1995), Rudolph (1977), Schraff (2004)	
Bob Hayes	M	1964	100 m	Yes	2	Hayes (1990)	
Wyomia Tyus	F	1964	100 m	Yes	7	Davis (1992), Carlson (1995), Tyus (2010)	
		1968	100 m				
Tommie Smith	M	1968	200 m	Yes	3	Smith & Steele (2007)	
Evelyn Ashford	F	1984	100 m	Yes	1	Davis (1992), Hornbuckle (1995b)	
Florence Griffith Joyner	F	1988	100 m, 200 m	Yes	2	Davis (1992), Hornbuckle (1995c)	
Carl Lewis	M	1984	100 m	Yes	NA	Lewis & Marx (1990)	
		1988	200 m				
			100 mb				
Ben Johnson	M	1988	100 mb	Yes	3	Christie (1988)	
Gail Devers	F	1992	100 m	Yes	6	Hornbuckle (1995d)	
		1996	100 m				
Gwen Torrence	F	1992	200 m	Yes	7	Hornbuckle (1995a)	
Michael Johnson	M	1996	200 m	Yes	5	Johnson (1996)	
Marion Jones	F	2000	100 mc	Yes	1	Jones (2004), Gutman (2000)	
Usain Bolt	M	2008	100 m	Yes	4	Bolt (2010), Cantor (2011), Irving (2010)	
		2012	200 m,				
			100 m,				
			200 m				
Notes.

a Defined as representing their country in international competitions (e.g., Olympic Games, World Championships).

b Awarded the gold medal because Ben Johnson was disqualified as winner for using performance enhancing drugs.

c Disqualified as winner for using performance enhancing drugs.

The biographies reported that adults (e.g., teachers, coaches) initially recognized the superior sprinting ability of nine sprinters (five women) and encouraged them to begin formal sprint training or competition. For example, the superior abilities of Wilma Rudolph, Helen Stephens, and Wyomia Tyus were discovered while they played basketball (Table S2), whereas Bolt (2010) and Hayes (1990) were discovered while they played cricket and baseball, respectively. In five cases (two women), sprinters reported recognizing their superior sprinting ability beginning in childhood. For example, Marion Jones reported that she was “always fast” and excelled at multiple sports (Jones, 2004) and Tommie Smith reported that he excelled at all sports as a schoolboy (Smith & Steele, 2007).

Sprinters required one to seven years of training to reach world class status (men: median = 4 years, mean = 4.6 ± 2.0 years, n = 7; women: median = 2.5 years, mean = 3.1 ± 2.4 years, n = 8; Table 1, Fig. 1). In fact, eight sprinters qualified for the Olympics as teenagers (Table S2).

Figure 1 The number of years of training required to reach world class status by male and female Olympic 100 and 200 m champions and the 20 fastest 100 m American male sprinters.

For 10 of 15 sprinters there was no evidence that they had participated in organized sports of any kind prior to the recognition of their superior sprinting ability or their initiation of deliberate sprint practice.

The results of this study clearly contradict the DPM: sprinters were consistently fast prior to formal training, achieved world class status in much less than ten years, and, in most cases, their exceptional development cannot be attributed to skill transfer. Nonetheless, this study has two possible limitations. First, the sample size of 15 is modest. Second, many of the individuals became Olympic champions several decades ago. Because world class sprint performances have continued to improve (Seiler, DeKoning & Foster, 2007), this raises the question of whether our results would differ if we used a more contemporary sample of sprinters. Study 2 was designed to address these limitations.

Study 2: Biographies of fastest 20 U.S. males of all time

In Study 2, we examined the development of the 20 fastest male U.S. 100 m sprinters. This is an excellent sample because the U.S. has been one of the strongest sprinting countries since the onset of modern international competition and record keeping (Lawson, 1997). This is revealed by the fact that 14 of 20 of these men won at least one individual World Championship or Olympic sprint medal (100 m, 200 m, or 60 m indoors); four of the others have won at least one relay medal at the World or Olympic championships. Moreover, all of these men achieved performances that meet contemporary standards of world class performance, including the 2012 Olympic A Qualifying Standard (i.e., 10.18 s automatically qualifying them to participate in the Olympic Games; http://www.usatf.org).

We again examined whether these sprinters were exceptional prior to initiating formal training and how long it took for them to reach world class status. We also searched for evidence indicating that these men were unexceptional relative to their peers prior to their beginning formal sprint training.

In addition, we documented the trajectories of performance improvement, particularly the percentage of improvement after age 19. The DPM makes no quantitative claim regarding the magnitude of improvement among regularly training adult athletes. However, the “talent matters” framework implies that once athletes have reached physical maturity and done some formal training, subsequent improvements will be relatively modest.

Methods

We used methods similar to those in Study 1 with the following two caveats. First, with the exception of Carl Lewis, book length biographies were not available for these athletes. We thus obtained information from magazines, newspapers, and internet sources. Second, we classified athletes as first reaching world class status upon first meeting either of the following criteria: (1) representing the U.S. in international competition (e.g., Olympic Games, World Championships, Pan American Games in an individual sprint event or as a member of a relay team) or (2) participating in the U.S. Olympic Trials which requires the athlete to meet Olympic A or B standards to qualify to compete at the Trials. Four of these athletes (Gatlin, Mitchell, Montgomery, Williams) were sanctioned for using PEDs at least once in their careers. Eight of the 20 sprinters (Bailey, Crawford, Dix, Gatlin, Gay, Padgett, Patton, Williams) competed in 2012 when we finished gathering data for this study. One athlete, Carl Lewis, was also included in Study 1.

We obtained information on athletes’ best performance at the age 19 from U.S.A. Track and Field (http://www.usatf.org), International Association of Athletics Federation (http://www.iaaf.org), or track and field historian Walter Murphy (pers. comm., 2011). We choose age 19 as a convenient cut-off age for comparisons between early and life-time fastest sprint performances because IAAF defines a Junior athlete as one who is 19 years of age or younger (http://www.iaaf.org). We obtained lifetime personal best performances from U.S.A. Track and Field (http://www.usatf.org). For these best performances, we only counted times that were legal (i.e., not wind-aided, wind less than 2 m per second).

In order to provide a more comprehensive picture of improvement, we plotted yearly best performances for the fastest 10 sprinters in this sample and plotted them as a function of age. We obtained data (though 31 December 2013) from the International Association of Athletics Federation (http://www.iaaf.org) and again only included legal times.

Results and discussion

We were able to obtain information regarding the development of 12 of 20 sprinters, and these data are summarized in Table 2. All 12 were recognized as exceptionally fast relative to their peers before or coincident with their initiation of formal training. There was no indication that any sprinter was initially unexceptional.

Table 2 Histories of the 20 fastest male American 100 m sprinters.

Ranking of sprinters and fastest 100 m times at age 19 and older obtained from U.S.A. Track and Field (http://www.usatf.org) and I.A.A.F. (http://www.iaaf.org) performance data.

Rank	Name	Year of
Birth	Age when
superior sprinting
speed first
recognized	Age at start
of DPc	Fastest time
at age 19	Fastest
time	Percent
improvementa	Years from
start of DP
to world
class statusb	Reference	
1	T. Gay	1982	13	13	10.27	9.69	5.64	9	Maloney (2007), Hendershott (2007)	
2	M. Greene	1974	8	8	10.19	9.79	3.93	13	Layden (1997), Hendershott (2000), Deford (2001)	
3	L. Burrell	1967	14	14	10.46	9.85	5.83	7	Hollobaugh (1991), Nooden (1991)	
4	J. Gatlin	1982	14	14	10.08	9.85	2.28	6	Hendershott (2005), Layden (2004)	
5	C. Lewisd	1961	NAe	NA	10.00	9.86	1.40	NA	Lewis & Marx (1990), Hurst (1994), Gleason (1980), Hendershott (1989)	
6	S. Crawford	1978	12	12	10.51	9.88	5.99	11	Denman (2006)	
7	W. Dix	1986	9	14	10.06	9.88	1.79	4	Landman (2008)	
8	R. Bailey	1989	15	15	10.28	9.88	3.89	5	Binder (2012)	
9	T. Padgett	1986	NA	NA	10.00	9.89	1.10	NA	Clemson University (2014)	
10	D. Patton	1977	NA	NA	NA	9.89	NA	NA	Patton (2014)	
11	D. Mitchell	1966	6	6	10.21	9.91	2.94	15	Hendershott (1994a), USATF (2000)	
12	L. Scott	1980	NA	NA	10.29	9.91	3.69	NA	USATF (2008)	
13	A. Cason	1969	NA	NA	10.08	9.92	1.59	NA	Hendershott (1994b), IAAF (2014)	
14	J. Drummond	1968	9	9	10.25	9.92	3.22	14	Weiss (1991), Reid (1999)	
15	T. Montgomery	1975	NA	NA	10.11	9.92	1.88	NA	Hendershott (2002), Abrahamson (2003), Fish (2009)	
16	T. Harden	1974	NA	NA	10.32	9.92	3.88	NA	USATF (2001)	
17	C. Smith	1961	NA	NA	10.17	9.93	2.36	NA	Martin (1980), Lee (1987)	
18	M. Marsh	1967	11	11	10.22	9.93	2.84	8	Hendershott (1993), USATF (1997)	
19	I. Williams	1985	16	16	10.29	9.93	3.50	7	Ainsworth (2012)	
20	B. Williams	1978	16	16	10.45	9.94	4.88	4	Satterfield (1997), Korth (2000)	
Notes.

a Percent improvement = 1 − (fastest 100 m–100 m time at age 19) × 100.

b World class status defined as either (1) representing the U.S.A. at international championships (e.g., Pan American Games, World Championships, Olympic Games) in an individual sprint event or as a member of a relay team or (2) participating in the U.S. Olympic Trials which requires the athlete to meet Olympic A or B standards to qualify to compete at the Trials.

c DP = deliberate practice.

d C. Lewis was the 5th ranked long jumper in the world and 2nd ranked long jumper in the U.S.A. by the age of 18.

e NA = no data/information available.

In nine cases, adults reportedly first recognized a sprinter’s talent. Leroy Burrell (Hollobaugh, 1991) and Bernard Williams (Satterfield, 1997) were discovered while they played baseball and basketball, respectively, whereas track coaches identified the superior abilities of the other seven. In the cases of Carl Lewis and Walter Dix, their parents were the track coaches (Lewis & Marx, 1990; Landman, 2008).

Eight of the 12 sprinters for whom relevant data were available required less than 10 years of deliberate practice to achieve world class status (median = 7.5 years; mean = 8.7 ± 3.8; Table 2, Fig. 1).

Fastest 100 m times at age 19 were available for 19 of the sprinters (Table 2). They showed only modest improvement between their fastest time at age 19 and their personal record (mean improvement = 3.3 ± 1.5%; Table 2). They typically achieved their fastest time in their mid-20 s (median = 24.8 years, 25.2 ± 2.6 years; Table 2).

The trajectories of 100 m performance improvement as a function of age are displayed in Fig. S1. These show, both individually and collectively, that sprinters’ abilities generally improve from their late teens until their mid-twenties and then gradually decline. Presumably, the improvement generally reflects physical maturation and training and the decline reflects senescence. These trajectories must also be affected by other factors, such as motivation, injuries, racing conditions, and the use of performance enhancing drugs.

One concern about Study 1 and Study 2 is that 10 years might not have been necessary to achieve expertise for many sprinters because PEDS accelerated their development. This issue warrants consideration, but, for several reasons, the use of PEDs cannot provide a genuine defense for the DPM. First, some sprinters in Study 1 performed before the PEDS believed to substantially help sprinters (e.g., anabolic steroids) would have been available to them. It is thought that weightlifters and bodybuilders in East Germany, the USSR, and the USA first used anabolic steroids in the 1950s (Ungerleider, 2001; Yesalis, Courson & Wright, 2000). Anabolic steroids did not become widely used by track and field athletes until after the 1960 Olympics (Yesalis, Courson & Wright, 2000). Thus, PEDs seem unable to explain the rapid development of Jesse Owens, Helen Stephens, Wilma Rudolph, and Bob Hayes. Second, the biographies of Ben Johnson and Marion Jones indicated they began using PEDs after they had achieved world class performances. These athletes and their coaches acknowledged that PEDs allowed them to run faster, but stated that the gains, although certainly meaningful in allowing them to beat their competitors, were proportionally modest. At the 1989 Canada Commission of Inquiry into the Use of Drugs and Banned Practices Intended to Increase Athletic Performance, Ben Johnson’s coach, Charlie Francis, testified, “It’s pretty clear that steroids are worth approximately a meter [in the 100 m] at the highest levels. He [Ben Johnson] could decide to set up his starting blocks at the same line as all the other competitors, or set them up a meter behind them all” (Nooden, 1989). A one meter benefit from steroid use is equivalent to 0.1 s in a 10.0 s 100 m sprint. Similarly, recent admissions by Tim Montgomery (see Table 2) indicate that he reached world class status prior to using PEDs and that the performance benefits were proportionally modest, roughly 2–3% (Axon, 2013). Thus, PEDs seem unable to provide a plausible explanation for the rapid attainment of world class status by these sprinters.

The results of Studies 1 and 2 contradict the DPM’s predictions, but they have two plausible limitations with regards to initial performance. First, perhaps the initially exceptional running of elite sprinters does not represent sprinting talent specifically. For example, a child with more overall athletic experience than its peers, or one who physically matures earlier, might be exceptional in almost all areas, and this early success could be a precondition for later pursuing and excelling in various sports. Second, perhaps sprinters desire to portray themselves as unusually talented and therefore provide false accounts of their abilities. Study 3 was designed to address these limitations.

Study 3: Surveys of collegiate sprinters

In Study 3, we recruited individual sprint qualifiers for the 2012 National Collegiate Athletics Association (NCAA) national championships to complete an online survey. We asked sprinters about their speed relative to their peers as children and adolescents. To address the specificity of their athletic ability, we also recruited a control group, collegiate throwers (e.g., shot put, discus, javelin) who qualified for these meets. The “talent matters” framework predicts that sprinters generally will recall being faster than their peers as children and adolescents than will the throwers. To further address specificity, we also asked about physical strength and overhand throwing ability. We predicted that throwers would recall being stronger and having better overhand throwing ability as youths than would sprinters.

The surveys also allowed us to obtain systematic data on sprinters’ performances in their first season of high school competition, which was generally coincident with their onset of formal training. Again, the “talent matters” framework predicts that sprinters will be much faster than most of their peers even at this early stage in their careers, whereas the DPM does not.

Methods

Ethics statement

The Chair of the Human Research Review Committee at Grand Valley State University reviewed the study protocol (Protocol 338194-1) and certified it as approved and exempt from full committee review.

Recruitment of subjects

We attempted to recruit all male and female individual qualifiers in the 100 m, 200 m, and 400 m sprints and shot put, discus, and javelin throws from the 2012 NCAA Outdoor Track and Field National Championships; lists were available online (http://www.ncaa.com). We recruited individuals from Divisions I, II, and III. The Divisions reflect, on average, the financial commitments made by colleges and universities to their athletes. Division I includes the largest athletic programs that provide the most athletically related financial aid for student-athletes, Division II institutions provide athletes limited financial aid, and Division III institutions do not provide athletically related financial aid (http://www.ncaa.org). Consequently, the most accomplished athletes (e.g., fastest sprinters) typically attend Division I institutions whereas the least accomplished generally attend Division III institutions. NCAA institutions are almost entirely comprised of U.S. schools.

We searched for email addresses through each school’s online directory and emailed all whom we could. In cases where we could not find email addresses, we attempted to make contact via Facebook. We were able to contact 72 of 114 candidate male sprinters (DI, n = 57; DII, n = 38; DIII, n = 19), and 72 of 146 female sprinters (DI, n = 59; DII, n = 42; DIII, n = 45). Of those contacted, 7 males (10%) and 13 females (18%) participated. In a similar manner, we attempted to contact all male and female individual qualifiers for the championship meets in the shot put, discus, and javelin throws. We were able to contact 83 of 159 male throwers (DI, n = 68; DII, n = 42; DIII, n = 49), and 107 of 169 female throwers (DI, n = 63; DII, n = 47; DIII, n = 59). Of those contacted, 18 males (22%) and 26 females (24%) participated. Numbers of qualifying athletes in each Division vary because some athletes qualified for multiple events and the number of athletes that met each Division’s championship qualifying standards varied.

The initial recruitment statement requested individuals to participate in a survey study of the “Development of elite athletic ability.” Individuals were informed that they had been contacted because they had qualified for the 2012 NCAA Outdoor Track and Field Championships. They were informed that the survey would take 5–10 min to complete and could be accessed by following an embedded link. No incentives for participation were offered. We first solicited responses from athletes from 13–15 July 2012, and this yielded 35 responses; we solicited responses again on 29 July 2012, and this yielded 29 additional responses.

Survey

The survey was implemented with the commercial platform SurveyMonkey. It began with the item, “To the best of your recollection, how would you compare your SPRINTING SPEED to others your own age and gender when you were 6–10 years old?” Five choices were offered, “much slower”, “slower”, “about the same”, “faster”, and “much faster”. The next item was the same except that the age range was 11–15 years old. Then, for each age range, there were similar multiple-choice items addressing physical strength and overhand throwing ability. We chose these age ranges because (a) 6–10 years constitutes a range before the typical onset of puberty and an age range when children are in school and can compare their athletic abilities (e.g., sprinting and throwing) with a larger group of peers than was available to them before attending school and (b) 11–15 years captures the onset of puberty (Jones & Lopez, 2006) but is earlier than most elite sprinters in Studies 1 and 2 reported, or were reported, to have begun formal sprint training with coaches.

The survey also included the following items:

• “If you competed in any of the following individual events in your FIRST YEAR OF HIGH SCHOOL track and field, please report your best performance in the event(s) during this FIRST YEAR OF HIGH SCHOOL track and field”. This was followed by a list of all common track and field events and a text box for each.

• “How old were you or what grade were you in at the end of YOUR FIRST SEASON OF HIGH SCHOOL track and field?”

• “To the best of your recollection, at what age (or grade) did you begin to seriously concentrate on track and field? (By seriously concentrate, we mean giving much attention and effort to training, usually with a coach.)”

The questionnaire also included items addressing gender, age, receipt of athletic-related financial aid, level of competition (e.g., Division I, II, or III), sports played prior to college besides track and field, recollections of first timed race, and best lifetime performances in all track and field events. No individually identifying information was sought, such as name or school.

Normative data

Normative data are required to assess the initial sprinting performance of elite sprinters. Because surveys (see below) indicated that these sprinters generally began regular training in 9th or 10th grade (usually ages 14–16) and usually reported best times for their first high school season, we focused on this age, and used two approaches to estimate normative data. First, we extrapolated 100 m, 200 m, and 400 m times (standard distances in U.S. high school meets) from normative values of 50 m times for a large representative sample of 15 year-old Australian schoolchildren (Catley & Tomkinson, 2013). We used Australian data because we could not find data from the U.S., and we have no reason to suspect that athletic abilities of the children from these nations differ substantially. We multiplied normative 50 m times by 2 to obtain 100 m benchmarks and by 4 to obtain 200 m benchmarks; because even world class runners slow by at least 10% when running 400 m, we multiplied 50 m times by 8.8 to obtain 400 m benchmarks. Thus, for females, 50th percentile benchmarks were 17.2 (100 m), 34.4 (200 m), and 75.7 s (400 m); 95th percentile benchmarks were 15.4, 30.8, and 67.8 s. The corresponding benchmarks for males were 15.4, 30.8, and 67.8 s (50th percentile), and 14.0, 28.0, and 61.6 s (95th percentile).

This method of determining benchmarks is conservative because our examination of high school data (see next paragraph) shows that children slow with increasing sprint distances even for 100 m and 200 m distances. In other words, if we had used more realistic, but difficult to determine, benchmarks, the high school performances of the collegiate sprinters would seem even more exceptional.

Our second approach to establishing the relative abilities of the sprinters focused on the upper boundary of performance. We did this by documenting the fastest 100 m and 200 m times recorded by 9th or 10th graders at high school divisional championship meets held in 2012. To obtain a reasonably representative sample, we first identified a website with track and field results for most U.S. high schools (http://www.athletic.net/). We searched 10 U.S. states in alphabetical order, looking for the first high school in alphabetical order in each state with results from the 2012 season. We focused on this school’s meet prior to the state championship meet, which was generally called a conference, sectional or division meet. These meets included 4–16 teams (median = 9.5) and would be open to all or nearly all pupils at each school. The mean school population (9th–12th grade) at each divisional meet ranged in size from 280 to 2100 students (median = 1483). Thus, the fastest 9th or 10th grade performances would generally represent the fastest male and female in a population of roughly 2,000–5,000 peers of the same sex and age. The median fastest times among 9th and 10th grade female performers were 12.96 and 26.45 s. For males, the median fastest times for 9th and 10th graders were 11.41 and 23.25 s. We consider these times to indicate performance at the 99th percentile or greater. We did not include median best 400 m times because many of these meets did not include a 9th or 10th grader among their finalists.

Results and discussion

Contrary to the DPM, collegiate sprinters recalled being faster relative to their peers than did collegiate throwers (Table 3; Fig. 2). This difference was significant and substantial for recollections of 6–10 and 11–15 years of age, and the differences held within men and women (Table 3). In fact, 90% of sprinters reported they were faster or much faster than their peers at 6–10 years of age and 80% reported they were faster or much faster at 11–15 years of age. As we predicted, throwers recalled being stronger and having better overhand throwing ability relative to their peers than did sprinters, and these differences held robustly for both age ranges and within men and women (Table 3). These results corroborate Studies 1 and 2 by showing that expert sprinters consistently recalled being faster than their peers as children. Furthermore, these recollections were at least somewhat specific to sprinting and so cannot be dismissed as a manifestation of general athletic ability.

Figure 2 The recollections by Division I, II, and III qualifiers for the 2012 National Collegiate Athletic Association (NCAA) Outdoor Track and Field Championships of their sprinting, strength, and over-hand throwing abilities as youths relative to their peers.

Relative ability: 5 = much faster, stronger, or better; 4 = faster, stronger, or better; 3 = about the same; 2 = slower, weaker, or worse. Mean relative ability plus one standard error of the mean is illustrated for each category.

Table 3 Recollections of childhood and adolescent athletic abilities of sprinters and throwers who qualified for the 2012 U.S. collegiate track and field outdoor championships.

Ability		Sprinters (n = 20)	Throwers (n = 44)	df a	t b	Cohen’s d	
		Mean	SD	Mean	SD				
6–10 years									
Sprinting									
	Total	4.3c	0.8	3.5	1.2	62	2.72**	0.79	
	Men	4.3	1.1	3.3	1.3	23	1.78	0.83	
	Women	4.2	0.6	3.6	1.0	37	2.00	0.73	
Strength									
	Total	3.3	0.8	3.9	0.8	62	3.21**	0.86	
	Men	2.7	0.5	3.9	0.9	23	3.27**	1.64	
	Women	3.5	0.8	3.9	0.7	37	1.58	0.52	
Throwing									
	Total	3.1	0.7	4.2	0.9	61	4.80**	1.23	
	Men	3.1	0.4	4.0	1.0	23	2.25*	1.17	
	Women	3.0	0.8	4.3	0.9	36	4.32**	1.49	
11–15 years									
Sprinting									
	Total	4.2	0.7	3.2	1.0	62	3.88**	1.10	
	Men	4.0	0.8	3.1	1.0	23	2.22*	1.04	
	Women	4.2	0.7	3.3	1.0	37	3.07**	1.10	
Strength									
	Total	3.5	0.9	4.3	0.8	61	3.22**	0.87	
	Men	2.7	0.5	4.1	1.0	23	3.37**	1.73	
	Women	4.0	0.7	4.4	0.6	36	1.76	0.58	
Throwing									
	Total	3.1	0.7	4.6	0.6	62	8.63**	2.25	
	Men	3.0	0.6	4.5	0.7	23	4.98**	2.31	
	Women	3.2	0.8	4.7	0.5	37	7.06**	2.24	
Notes.

* p < 0.05.

** p < 0.01.

a Degrees of freedom differ because some participants did not complete all items.

b Student’s t-test.

c Values in table represent scores on surveys on a five point scale with higher scores indicating higher self-rated ability; see Study 3 Methods for scales.

Table 4 provides information for each sprinter regarding their background, onset of training, and best performances. Seventeen of 20 sprinters reported at least one best performance in their first season of high school competition, and only two of these reported they had begun serious training prior to this. Of the 15 sprinters who reported first season high school performances and no prior serious training, 13 of 15 were age 15 or younger at the end of this first season, supporting our decisions regarding age-appropriate benchmarks (see Methods). All 27 performances recalled by these 15 sprinters were faster than 95th percentile benchmarks. Moreover, seven of these sprinters recalled at least one performance faster than the 99th percentile benchmarks, and two of the others recalled performances that were within 0.5 s of 99th percentile benchmarks. These results represent more objective evidence that, relative to their peers, these sprinters were exceptional prior to the accumulation of substantial training.

Table 4 U.S. collegiate sprinters’ recollections of their onset of training and best performances.

Sex	NCAA
division	Age	Began
training	Age first year
high school	First year high school best performance	Lifetime best performance	
					100 m	200 m	400 m	100 m	200 m	400 m	
F	I	18	*17a	14	12.7	26.3	NAb	NA	NA	51.1	
F	I	22	18	14	NA	NA	59.Xc	11.4	23.4	56.X	
F	I	24	*21	*17	NA	NA	NA	11.1	23.1	NA	
F	II	23	15	14	NA	26.X	58.X	NA	24.1	53.0	
F	III	21	*13	14	13.X	28.X	NA	11.9	26.0	62.2	
F	III	22	*15	15	12.7	27.2	NA	12.0	25.4	NA	
F	III	23	*19	*15	13.1	26.X	60.0	NA	25.3	56.4	
F	III	20	*16	*15	13.5	26.8	59.9	12.4	24.5	55.5	
F	III	22	*17	*17	13.X	27.X	59.X	12.0	24.3	55.2	
F	III	21	*16	*15	13.X	27.X	61.X	13.X	26.0	56.8	
F	III	20	*13	*14	15.X	27.X	62.X	12.4	24.6	58.1	
F	III	19	*15	14	NA	28.X	63.X	NA	24.9	54.8	
F	III	21	18	NA	NA	NA	NA	12.9	26.0	56.5	
M	I	20	*15	14	11.0	22.5	52.9	10.2	21.1	49.2	
M	I	20	18	*16	11.3	23.5	53.4	10.6	20.9	46.0	
M	I	20	*16	*15	11.2	23.5	54.X	10.9	21.2	46.3	
M	II	22	NA	NA	NA	NA	NA	NA	21.5	47.1	
M	II	21	*17	15	11.3	22.8	49.9	10.6	21.1	46.1	
M	III	22	14	14	NA	NA	57.X	10.6	21.3	48.8	
M	III	21	17	*16	NA	22.2	49.4	NA	22.1	47.4	
Notes.

a * Indicates age estimated from reported grade (e.g., 9th grade = 15 years).

b NA, no data provided by athlete.

c Performances including an “X” after the decimal indicate uncertainty about exact time.

A possible limitation of Study 3 is that the response rate of college athletes was low. However, a low response rate is reasonable because we did not provide athletes with incentives to participate and because we attempted to contact them after the academic year had ended. More importantly, to minimize response bias, we constructed the survey questions to appear neutral to the DPM or the “talent matters” framework.

General Discussion

The three studies of sprinter development in this paper focused on testing two crucial predictions of the DPM. We begin our discussion by considering each prediction. We then examine the implications of our findings.

Elite sprinters are initially remarkable

The first DPM prediction is that elite sprinters should have generally been unremarkable prior to training. Contrary to this, the biographical materials examined in Studies 1 and 2 indicated exceptional initial ability for all 26 world class sprinters for whom we were able to obtain relevant information. Study 3 corroborated this pattern in national qualifying collegiate sprinters, showing that they recalled being faster or much faster than their peers as children. In addition, these collegiate sprinters reported achieving performances in their first season of high school competition that would have exceeded 95–99% of their peers despite the fact that most had begun formal training that same season.

A limitation of these studies is that the use of biographical materials relies on the retrospective recall of information from many years earlier, and this information may be inaccurate or biased (Shiffman et al., 1997), although studies have demonstrated moderately high correlations between information obtained by retrospective recall and that found by examining diaries (Ericsson, Krampe & Tesch-Römer, 1993; Sloboda et al., 1996; Baker, Côte & Deakin, 2005; Ward et al., 2007). Most empirical studies addressing the DPM framework use systematic methods, such as requiring participants to maintain regular training diaries (Ericsson, Krampe & Tesch-Römer, 1993; Sloboda et al., 1996; Baker, Côte & Deakin, 2005; Hodges et al., 2004). Such studies have not been conducted for sprinting, and they seem impractical. This is because, to our knowledge, there are no formal training programs or sports academies that endeavor to train “typical” children or adolescents so that they develop into elite sprinters. Apparently formal, dedicated sprint training is only taken up by individuals who are recognized as being exceptionally fast prior to formal training.

We also note that there are several reasons why inaccurate or biased biographical materials cannot provide a satisfying explanation for our results. First, if many sprinters in Studies 1 and 2 were not exceptionally fast prior to formal training, it would seem that at least some coaches, competitors, or peers would attempt to report the truth to biographers and journalists. For example, if Gwen Torrence and Evelyn Ashford had not, as untrained high school students, each beaten their school’s star (male) football player in a race, as their biographies attested (Hornbuckle, 1995a; Davis, 1992), we might have expected someone to dispute these or the many similar claims in other biographies. Similarly, most biographical accounts of extraordinary youth sprinting ability are corroborated by publicly documented timed performances. Moreover, objective facts such as race results are more accurately recalled than are subjective states (e.g., recalling the amount of effort put into a particular practice session) (Brewer, 1998).

A second point is that Study 3, based on national qualifying collegiate sprinters, fully corroborated Studies 1 and 2. Although it was based on self-reports, the responses were anonymous, and this should have minimized self-presentation bias that might have occurred in the biographies (e.g., champion sprinters desiring to portray themselves as being innately gifted.)

A final point is that DPM proponents have used biographical materials similar to ours to support their key claims. Most notably, Ericsson, Krampe & Tesch-Römer (1993) and Howe, Davidson & Sloboda (1998) reviewed biographies and retrospective studies of a variety of eventual experts (e.g., musicians, painters, chess players) and argued that these indicate that the experts’ initial performances were consistently unexceptional and that many years of deliberate practice always preceded their emergence as experts. Studies 1 and 2 demonstrate beyond doubt that this pattern does not hold for the biographies of most (and perhaps all) expert sprinters. There is no reason to dismiss sprinters’ biographies as highly inaccurate or biased while accepting the veracity of other biographies.

Our studies are also notable because they ruled out two alternative explanations for sprinters’ initially exceptional abilities. In Study 1 we addressed the transfer hypothesis, whereby remarkable initial performance in one domain, such as sprinting, might be due to previous training in another, such as football or baseball (Howe, Davidson & Sloboda, 1998). We showed that a transfer hypothesis is not viable because the biographies of Olympic champions revealed that two-thirds of them did not participate in organized sports prior to beginning sprinting. A DPM proponent might protest that perhaps informal sports participation was crucial for sprinting development. This argument has some validity: informal (and difficult to measure) experience (e.g., play) might be crucial. However, accepting this argument would still entail abandoning the underlying premise of the DPM, that expertise must be based on formal, deliberate training.

With Study 3 we addressed the possibility that remarkable initial sprinting ability might be merely indicative of unusual general athletic ability or early physical maturation. Contrary to this, the collegiate sprinters generally recalled being faster or much faster than their peers as children, whereas another group of similarly elite athletes, throwers, did not recall being as exceptionally fast. That the opposite pattern occurred for physical strength and overhand throwing shows that sprinting ability is not merely a manifestation of general athletic ability; it is specific, at least to some extent, consistent with definitions of talent (Howe, Davidson & Sloboda, 1998). As we will discuss below, however, speed is crucial for many other sports so it is not surprising that many champion sprinters excelled in sports besides track and field.

There are other possible concerns about our tests of this prediction, but none seem compelling. One concern is that, although we found evidence of initially outstanding speed in all three studies, the sample sizes were modest (i.e., n = 26 for Studies 1 and 2 combined; n = 20 sprinters for Study 3). This does not seem like a major weakness, however, because genuine experts are, by definition, rare, which is why many DPM studies use modest samples. For example, the seminal paper by Ericsson and colleagues (1993) included data from only 54 musicians, only 22 of whom were considered expert.

A second, related concern is that our results are based on expert sprinters and perhaps may not apply to the development of sprinting abilities in a general population of athletes. In other words, for most athletes initial performance might explain little or no variation in the attainment of sprinting ability. This point is largely valid: studies of the initial performance and development of sprinting ability with proper training in broad range of individuals are certainly desirable. Nonetheless, this point does not mitigate the challenge our results pose to the DPM. This is because DPM proponents have consistently stressed the study of genuine experts and they have, with no apparent exceptions, assumed that the same principles apply to all individuals, i.e., there is no meaningful distinction (e.g., talent) between expert and non-experts besides in their training (Ericsson, Krampe & Tesch-Römer, 1993; Howe, Davidson & Sloboda, 1998). In addition, we note that Study 3 included many athletes whose best performances to date are far from world class (Table 4); given the base rate occurrence of world class performances and the typical patterns of world class sprinter development (Fig. S1), most of these sprinters never will achieve world class performances. Nonetheless, their initial performances in childhood and high school were generally outstanding. This indicates that the patterns we documented do occur generally.

Elite sprinters break the “10 year rule”

Our studies also contradict the DPM’s prediction that at least 10 years of deliberate practice are necessary to achieve expert level performance. The results of Study 1 showed this because the median time to reach world class status for 15 Olympic champions was only three years. Study 2’s results indicated a median of 7.5 years to reach world class status for the fastest 20 men in U.S. history. This might seem roughly consistent with the 10 year rule, but it must be noted that our estimate of the initiation of formal training was highly conservative. For example, if someone began participating in track and field competitions at age 8 (Table 1), we considered that as the beginning of formal training. However, we found no indication that, as children, any of the sprinters in Study 1 or Study 2 engaged in anything remotely similar to the demanding, time-intensive training that has been documented for future elites in other sports such as tennis, swimming, and gymnastics (Bloom, 1985).

Moreover, similar to other previous studies (Starkes et al., 1996; Kalinowski, 1985; Helsen, Starkes & Hodges, 1998; Hodges & Starkes, 1996) our measure of the duration until reaching world class status confounds training with physical maturation. For example, if a child begins training at age 10 and reaches world class status at age 20, one interpretation is that 10 years of deliberate practice were necessary for this improvement. An alternative interpretation is that elite performance can only occur after physical maturation and that a talented individual could have reached the same performance level at age 20 if they had only begun training at age 18. Although this might seem far-fetched, in the course of our research we discovered two documented cases of men beginning formal sprint training as adults and reaching world class status within one year. The athletic biographies of these sprinters, Dave Sime and Delano Meriwether, are impossible to reconcile with the DPM, and we have summarized their biographies in Table S3.

Moreover, the biographies of Sime and Meriwether patently contradict the DPM’s claim that an athlete who starts deliberate practice at a relatively late age would not be able to “catch up” to an athlete who started training earlier (Ericsson, Prietula & Cokely, 2007; Baker, Cobley & Fraser-Thomas, 2009) and the complementary view that extensive experience in a sport is necessary to reach world class status (Williams & Ford, 2008). We also note that rapid development of sprinting expertise was not limited to men. Female Olympic gold medal winners Helen Stephens, Wilma Rudolph, Evelyn Ashford, and Marion Jones all achieved world class sprinting status within three years of beginning training (Table 1).

DPM proponents might argue that achieving expertise in less than 10 years might reflect extraordinarily intense training, so that, in a sense, 10 years of training might be compressed into nine (or fewer) years. However, the biographies of world class sprinters do not fit the claim of unusually intensive training (see Table S2). For example, Usain Bolt’s biographies (Cantor, 2011; Irving, 2010), including his autobiography (Bolt, 2010), document that his often extreme disinterest in training has been very frustrating for his coaches. In fact, Bolt was left off of the Jamaican national team competing at the 2003 IAAF World Championships because of his lackadaisical attitude (Cantor, 2011).

We also note that sprint training requires high intensity efforts on the track and in the weight room several days per week with the remaining days spent in rest and low intensity recovery exercises (Francis & Coplon, 1991; Ward & Dintiman, 2003; Smith, 2005; Edwards, 2012). Even Ericsson, Krampe & Tesch-Römer (1993) and Ericsson (2006) recognized that very intensive sprint training is difficult for long periods even with periods of rest. Consequently, the time that sprinters spend practicing sprinting may be very brief. For example, Carl Lewis, considered by many to be the paragon of professionalism as a track and field athlete, reported that a typical sprint training session consisted of sprinting 200 m six times averaging 23.0 s for 200 m with one minute of rest between sprints (Hurst, 1994). Keith Roberts, Grand Valley State University’s sprint coach, (pers. comm., 2013) estimates that collegiate and professional sprinters typically spend 600–700 h per year training on the track and in the weight room combined and that high school sprinters would typically spend considerably less time. At the professional rate, it would take between 14 and 17 years to accumulate 10,000 h of deliberate practice.

The DPM has been falsified for sprinting

Is there any way to reconcile our results with the DPM? We do not think there is. As reviewed here, several lines of evidence contradict the DPM, and even the data that seem consistent with it (e.g., some sprinters “needing” ten years of deliberate practice) are easily accommodated into the “talent matters” framework. Furthermore, in our review of biographical materials and discussions with coaches and experts, we did not encounter even one account of an elite sprinter that prototypically fits the DPM (e.g., a sprinter who was mediocre compared to a general population of their peers and who engaged in deliberate practice for many years and eventually became elite). The absence of such accounts, together with the data in our three studies and many complementary physiological and genetic studies (Bouchard et al., 2011; Costill, Fink & Pollock, 1976; Ahmetov & Fedotovskaya, 2012), allow us to conclude with confidence that the DPM has been falsified in the domain of sprinting.

Sprinting is an authentic expertise domain

Although the evidence indicates that the DPM has been falsified for sprinting, it might be argued that this does not seriously weaken the model because sprinting is an inauthentic or highly unusual expertise domain. One version of this argument is that sprinting is inauthentic because performances are highly constrained in that the goal is the same for every performance, to run as fast as possible. By contrast, prototypical domains, such as chess or music, demand far greater flexibility in decision-making and/or motor skills. This argument is unconvincing, however, because DPM researchers have frequently assumed the relevance of other highly constrained domains. For instance, Ericsson, Krampe & Tesch-Römer (1993) discussed how unremarkable adults can be trained to achieve prodigious performance in specific memory tasks, and Ericsson, Nandagopal & Roring (2009) noted that, with proper training, otherwise unremarkable adults can greatly improve their endurance running or complete thousands of pushups per day. Thus, it seems likely that if DPM proponents could demonstrate that most healthy adolescents can achieve outstanding sprint performances (e.g., <11.5 s for 100 m; see Study 2 and Table 3) with only a few months of deliberate practice, they would do so and cite this as evidence supporting the DPM. The fact that such a demonstration is apparently impossible must, therefore, count against the DPM.

A second possible argument is that sprinting is an inauthentic domain because it requires minimal skill. In other words, one might be dazzled by a professional pianist while dismissing a world class sprint performance as something that almost anyone could achieve, albeit more slowly. This argument is also unpersuasive because sprinting does require skill. Elite sprinters exert much effort in physical training (e.g., running, plyometrics, strength training) and technique (e.g., starts, transitions) (Francis & Coplon, 1991; Ward & Dintiman, 2003; Smith, 2005; Edwards, 2012). The resulting improvements may be modest (e.g., 1–6%; see Table 2), yet they can easily make the difference between being a mere qualifier for a championship meet and being the champion.

We also note that sprinting is one of the most popular sports across a broad spectrum of traditional and modern societies (Deaner & Smith, 2013; Gotaas, 2009; Guttmann, 2004a; Guttmann, 2004b; Sears, 2001). Sprinting’s popularity is also revealed by the global fame accorded to Olympic champions such as Jesse Owens (Baker, 1986), Wilma Rudolph (Smith, 2006), Bob Hayes (Hayes, 1990), Marion Jones (Jones, 2004), and Usain Bolt (Bolt, 2010). Furthermore, unlike prototypical expertise domains (e.g., music, chess), no special equipment is needed for sprinting, meaning that, in most societies, it can be undertaken by virtually any individual. These points suggest that sprinting is not merely an authentic expertise domain; it may be an ideal one.

Despite these points, development in other expertise domains, especially those based on decision-making or cognitive skills, could be substantially different than for sprinting and other domains of physical skill (Abernethy, Farrow & Berry, 2003). Nonetheless, there is mounting evidence that much individual variation in achievement in more cognitive expertise domains (e.g., music, chess, educational attainment) also cannot be accounted for by the DPM (Hambrick & Meinz, 2011; Meinz et al., 2012; Meinz & Hambrick, 2010; Hambrick et al., 2013; Wai, 2014).

Sprinting is a foundational expertise domain

We have argued that our results pose intractable problems for the DPM. However, the implications of these results are actually greater than showing that one authentic domain does not fit the model. This is because sprinting is a foundational domain in the sense that elite speed is necessary or at least highly advantageous in many other sports. Coaches and commentators frequently convey this in asserting that “speed kills” and much research supports the point (Huijgen et al., 2009; Little & Williams, 2005). For example, in American football, sprinting speed over 40 yards is a significant predictor of playing ability and highly recruited high school players often possess similar speed to professionals (Ghigiarelli, 2011).

The advantages of superior sprinting speed are particularly well illustrated in cases where elite sprinters take up new sports and reach world class status soon thereafter. For example, an Australian woman recruited to train for skeleton, a winter sliding sport, based on her 30 m sprinting speed reached world class status after only ten weeks of sport-specific training (Bullock et al., 2009). More recently, two American world class sprinters, Lolo Jones and Lauryn Williams, rapidly made the transition to world class bobsled competition. Within several months of taking up the sport, Jones helped the U.S. team win a world championship (http://sportsillustrated.cnn.com/more/news/20130228/lolo-jones-bobsled.ap/). Two years after beginning bobsled training, Jones was named to the 2014 U.S. Winter Olympics team. Her teammate Lauryn Williams, a 2012 Olympic gold medalist sprinter in the 4 × 100 m relay, was named to the 2014 Olympic bobsled team six months after beginning training (http://lauryn-williams.com/).

We also note that while this paper has focused on behavioral data revealing the existence of innate sprinting talent, complementary studies have implicated genetic and physiological mechanisms (Costill, Fink & Pollock, 1976; Macarthur et al., 2006). Studies using similar methods have indicated that characteristics crucial to success in other sports are also partly innate. For example, substantial heritability has been demonstrated for muscle mass and strength, critical contributors to success in power sports (e.g., American football, weightlifting) (Silventoinen et al., 2008; Stewart & Rittweger, 2006) and for untrained VO2max and VO2max responsiveness, which are vital for endurance sports (Bouchard et al., 2011; Bouchard et al., 2012). The convergence of this evidence suggests sprinting speed is only one of several innate ingredients that, depending on the particular sport, may be crucial for success.

Concluding Remarks

Our studies are the first to systematically show that: (1) a strong predictor, probably a precondition, for elite sprinting performance is exceptional speed prior to formal training, (2) this exceptional ability is at least partly specific to sprinting, and (3) many elite sprinters reach world class status in far less than 10 years, although they usually make modest improvements even after that (Fig. S1). Although these results are novel in the scientific literature, it is striking how closely they seem to correspond with folk wisdom or commonsense. In fact, in conversations about this research with colleagues, we found that everyone with at least some athletic experience anticipated our main findings.

Thus, the previous neglect of sprinting in the context of the DPM seems puzzling. On the one hand, it would seem that those skeptical of the DPM would have pointed out that sprinting strongly challenges it. The fact that they did not perhaps reveals the success of DPM proponents in steering research towards domains where the DPM is more difficult to rule out. On the other hand, the fact that DPM proponents neglected sprinting seems understandable: people are notorious for overlooking or discounting evidence that contradicts their views (Lord, Ross & Lepper, 1979; Munro, 2010).

Finally, we find remarkable the continued popularity of the DPM despite its empirical weaknesses and theoretical implausibility (Abernethy, Farrow & Berry, 2003; Hambrick & Meinz, 2011; Tucker & Collins, 2012; Ackerman, 2013; Detterman, Gabriel & Ruthsatz, 1998). We speculate that the model’s popularity reflects a more general desire to adhere to a “Blank Slate” view of human nature, whereby behavior is wholly shaped by the environment and that individuals have no inborn predispositions or talents besides the general ability to learn (Pinker, 2002; Tooby & Cosmides, 1992). Although contradicted by evolutionary theory and abundant empirical data, the Blank Slate remains popular, apparently because of its supposed benevolent consequences (e.g., that anyone can achieve expertise) (Pinker, 2002; Detterman, Gabriel & Ruthsatz, 1998).

However, the Blank Slate view can have negative impacts as well. As an example, imagine that a youngster expressed a strong passion to become a world class sprinter and they trained with great dedication under expert coaches for 10 years. Our studies indicate that, despite this training, the individual will not realize or even approach their goal unless they happen to possess extraordinary talent. The DPM view, by contrast, logically implies that their inability to become world class must be solely due to training failures on the part of the individual or their coaches. We believe that such blame would be unwarranted and undesirable and that a realistic view—that both training and talent are necessary—is preferable.

Supplemental Information

Tables S1--S3 Supplemental tables

Click here for additional data file.

Figure S1 Figure S1

Click here for additional data file.

We benefited from conversations with Rick Albrecht and John Kilbourne and the GVSU track and field coaches, especially Jerry Baltes, Blaine Maag, and Keith Roberts. Brian Gurta helped in the initial stages of data collection. Shadie Emiah helped find contact information for collegiate athletes. Walt Murphy provided important biographical information about sprinters. Zach Hambrick, Shane Mueller, Ross Tucker, Bo Winegard, and Mike Wolfe provided useful comments on previous versions of the manuscript as did two anonymous reviewers.

Additional Information and Declarations

Competing Interests

Author Contributions

Human Ethics

The authors declare there are no competing interests.

Michael P. Lombardo and Robert O. Deaner conceived and designed the experiments, performed the experiments, analyzed the data, contributed reagents/materials/analysis tools, wrote the paper, prepared figures and/or tables, reviewed drafts of the paper.

The following information was supplied relating to ethical approvals (i.e., approving body and any reference numbers):

The Chair of the Human Research Review Committee at Grand Valley State University reviewed the study protocol (Protocol 338194-1) and certified it as approved and exempt from full committee review.

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
