# Peer review of "You can’t teach speed: sprinters falsify the deliberate practice model of expertise"

_PeerJ, doi:10.7717/peerj.445_

## Round 0.1 · original submission · Minor Revisions

Michael,

I have read the manuscript and find it well written, impeccably referenced, and providing an interesting approach to falsifying the DPM (retrospective historiographic analysis). The DPM has certainly gained substantial popularity in popular press, and fueled by its non-intuitive premise, has become a part of the doctrine of cognitive psychology that researchers seem to question. It seems very useful to establish some of its boundary conditions, and perhaps chip away at its entire premise.

I have received reviews from two independent experts in this domain. Furthermore, I have experience in studying expertise in cognitive domains and studies of elite athletes, and so I have some background in this area. Reviewer 1 cautions about sample size, and would like to see additional data. Reviewer 2 is generally positive, and makes some nice suggestions for improving the manuscript. I think they both have valid points that should be considered. I think consistent with the PeerJ guidelines, I will ask for some minor revisions (see below), but am generally positive about the manuscript. With a topic as controversial as this, I think you should make efforts to really establish clearly which aspects of DPM are being falsified by the data, because if your claims are overly broad, it I think a very clear case needs to be made about what is being falsified and what is being supported, and revisions should definitely focus on exactly what is being falsified (more on this later).

But first, let me give my perspective on sample size here. Sampling all depends on the population you are trying to generalize to, and consequently, for studies 1 and 2, you are essentially sampling the entire population. In fact, in these cases, inferential statistics like t-tests aren't even really appropriate, although I'm not clear what the alternative is. For study 3, you are attempting to generalize to the entire population of qualifiers (a couple hundred) based by sampling them ALL, but only getting responses from 1/5. I think that it is rare for a psychological study to sample even 1% of the population it wants to generalize to, and so to some extent the sample size issue is not relevant for study 3 either. Nevertheless, the results are based on just 20 sprinters, and so the evidence is a bit limited Given the design of study 3, I don't think it is reasonable to try and re-solicit responses (and anyway, this is inconsistent with the PeerJ mandate) and I do think the data stands on its own, so I would be happy accepting the paper without collecting more data in that study

I am concerned that the presentation of Experiment 3 is somewhat confusing. Overall, I found it difficult to figure out what the study is really trying to establish. In the least, the table summarizing the results needs to be labeled better--it is unclear what many of the numbers even refer to. Please make efforts to clarify the results, tables, and figures for this experiment, as it is the central result of the study. I'd also like to point out that throwers may not all be appropriate; for example, javelin is only competed as a high-school sport in two states, and so no collegiate javelin throwers would likely have more than 4 years experience, which may limit results.

I'd like you to address several issues related to the strength of the conclusions that can be drawn from the presented data. The data are strong for what they are, but they do not deal with everything about the DPM. Here are some of my comments:

One counter to the argument offered here is that "Elite sprinters are faster to start with, but they still improve as they practice for the first decade". This is certainly consistent with DPM, although it might not be a strong interpretation. Yet I don' t think your data address this directly. Although the best sprinters may have started better, they may continue to improve. Studies 1 and 2 only look at the time to become 'world-class'. Yet these sprinters may have continued to improve. As an example, Carl Lewis got progressively faster from age 20 to age 30, after which he declined (see
http://www.iaaf.org/athletes/united-states/carl-lewis-1622#progression
), as did Tyson Gay (http://www.iaaf.org/athletes/united-states/tyson-gay-185464#progression) I would suggest augmenting study 2 to include an analysis of the progression across their career/age of at least a handful of these sprinters. I think you would find that they do tend to continue to improve over a decade of performance, which is an interesting caveat to both DPM and your thesis. I'd also suspect the improvement and eventual inflection is more tied to age than it is to years of practice. Showing such improvement over time helps illustrate one of the difficult confounds in expertise research--just because you continue to improve with practice does not mean that practice is what got you there, and this time-progression data establishes that the skill is indeed mutable with practice even among the best in the world, possibly countering the a criticism that sprinting shouldn't count . Finally, by marking some anchor points, (i.e., olympic qualifying times, a historic world record, etc.) you could easily establish that even though they improved, they started out as elite.

Another big concern related to the DPM is that the samples examined focus on those at the top. While this does establish a development path for the best sprinters, there might be entire classes of very good sprinters who are not at this level but by all rights are considered experts (i.e., repeatable superior performance), and these 'normal' sprinters might follow a DPM path. It might be true that, in general, a sprinter takes ten years to get to the top of their game, but the elite sprinters are born at the top. Focusing on the elites does not establish that sprinting in general falsifies the DPM, only that the most elite sprinters do. Unfortunately, there probably doesn't exist consistent time records for sub-elite sprinters (although perhaps as one reviewer suggested, looking at football receivers might be a good comparison). If you can find data to address this, that would be great; if it is not available, you need to address the fact that it is elite sprinters that falsify the DPM, and not sprinting in general.

I have a third concern that you have dealt with fairly well, but it is important to be sure it is clear. You have addressed the concern that one might offer that sprinting shouldn't really count, for whatever reason. This potential criticism is a form of what I have heard referred to as the 'no true scotsman' criticism, and a supporter of DPM can simply say "sprinting isn't what we mean when we say expertise". And maybe it isn't--sprinting certainly has mental and psychological components (especially the initial reaction time), but it must involve physical superiority. This is different from even middle-distance and long-distance running, which involve grappling with huge mental barriers, and it is different from any 'skilled' sport, which are more akin to the examples typically used by DPM proponents. The only other similar sports might be other sprinting domains (speed skating, swimming), and maybe things like weigthlifting. It would be good to add discussion of cognitive versus physical skill in sport and how they may impact DPM. This might play into the research (or at least anecdotes) regarding heritability of speed outside of humans--I think it has been established how important bloodlines are for horse sprinters (i.e., Secretariat's 'large heart' gene), and it would seem remarkable if speed is heritable in animals but not humans. Anyway, if any additional evidence can be brought that illustrates DPM has considered things like sprinting as relevant, it should be brought forth.

In conclusion, I feel the manuscript could stand some minor improvements. Along with responding to the comments of the reviewers, I think Study 2 should be augmented with an timeline progression of times for a subset of athletes, that would help establish the extent to which sprinting is amenable to practice and thus partly consistent with DPM, even though it cannot explain the most elite performers.

Reviewer 1 ·

Basic reporting

The article appears sufficient in basic reporting.

Experimental design

The weaknesses of the design are very small samples in Study 1 and Study 2 and the small survey response rates in Study 3. Typically, these weaknesses would lead me to directly reject this manuscript on sample size limitations alone.

However, Anders Ericsson (primary proponent of the DPM) himself relies on small samples and case studies, because this is how he defines expertise. Taken in that context, these samples appear similar in size.

Another strength of the design is the replication across three different samples from different sources.

I wonder if the authors might be able to find another sample (perhaps all sprinters rather than just U.S. sprinters? or perhaps running backs or other samples that require speed?) to enlarge their overall sample size and see if findings replicate. I think if this was done, it would greatly strengthen the paper.

Validity of the findings

This appears appropriate.

Additional comments

You might want to add in the N for Study 3 in the abstract.

Working memory (or general intelligence) is also central to educational and occupational expertise, is highly heritable, and yet another area where the DPM proponents have steered focus away from. See, for example: http://www.sciencedirect.com/science/article/pii/S0160289613001268 as well as articles in the entire special issue of expertise in the journal Intelligence.

DPM is largely based on restrospective studies with small samples based on Ericsson's very strict definition.

Line 152: I wonder if "highly select" is more appropriate than "conservative"

Line 178: when you say the sample size is modest, you might want to point out that Ericsson himself uses small samples, and cite the typical N's in his studies of people. If Ericsson's N's match up to yours, then you have a reasonable defense for small samples.

Line 200: as I mentioned in a prior section, why not expand beyond just U.S. sprinters? Would this enlarge your sample size?

I think it is quite obvious, as you do, that a practice only view is quite ridiculous given our understanding of the wide importance of talent across so many domains. I found your study quite interesting.

·

Basic reporting

The article is well-written and timely. As far as I understand them, the article meets the standards listed at right.

Experimental design

The research question is clearly defined, and is relevant and meaningful. The Methods are described in sufficient detail, and the research was conducted in conformity to ethical standards.

Validity of the findings

The research reported in this article investigated the question of whether training history is sufficient to explain individual differences in athletic performance, and in particular, sprinting performance. Across multiple studies, using historiometric and survey methodologies, the authors found evidence that exceptional runners showed early signs of promise.

The results of this study are quite clear: "In all documented cases, sprinters were exceptional prior to initiating training, and most reached world class status rapidly (Study 1 median = 3 years; Study 2 = 7.5)." This finding flatly contradicts Ericsson and colleagues argument that it takes at least 10 years of deliberate practice to reach an elite level of performance, which Ericsson and his colleagues stated clearly in a 2007 Harvard Business Review article: "Our research shows that even the most gifted performers need a minimum of ten years (or 10,000 hours) of intense training before they win international competitions" (Ericsson et al., 2007). The results of this study falsify this claim.

A particularly nice feature of this research is that the authors were able to retrieve objectively (and publicly) verifiable performance data, i.e., running times from the participants in their sample from earlier points in their career. For example, they explain: "We obtained information on athletes’ best performance at the age 19 from U. S. A. Track and Field (www.usatf.org), International Association of Athletics Federation (www.iaaf.org), or track and field historian Walter Murphy (pers. comm., 3 April 2011)." So, they weren't simply relying on retrospective reports.

Generally speaking, the results of this study are contrary to the view (Ericsson et al., 1993) that exceptional performance is largely attributable to "deliberate practice" and instead suggests that pre-training characteristics contribute to achieving a high level of performance in a domain. I believe that the findings of this study will make a nice contribution to the literature on expertise, and will be of interest to a broad audience.

Additional comments

To substantiate the point that proponents of the deliberate practice view have claimed that deliberate practice largely accounts for individual differences in performance, I suggest that the authors include the following quotation from Ericsson et al. (1993). (The authors may include this quotation in the Table S1 to which they refer in the text, but I can't seem to locate that table. However, I think this would be a good quotation to include in the main text. Or, instead of a supplementary table, this could be a main table in the article.)

"Because of the high costs to the individuals and their environments of engaging in high levels of deliberate practice and the overlap in characteristics of deliberate practice and other known effective training situations, one can infer that high levels of deliberate practice are necessary to attain expert level performance. Our theoretical framework can also provide a sufficient account of the major facts about the nature and scarcity of exceptional performance. Our account does not depend on scarcity of innate ability (talent) and hence agrees better with the earlier reviewed findings of poor predict-ability of final performance by ability tests. We attribute the dramatic differences in performance be-tween experts and amateurs-novices to similarly large differences in the recorded amounts of deliberate practice." (Ericsson et al., 1993, p. 392).

Also, to substantiate the point that proponents of the deliberate practice view have argued that a minimum of 10 years, or 10,000 hours, of deliberate practice is necessary to reach an elite level of performance, I suggest that the authors include this quotation from Ericsson et al. (2007), Harvard Business Review:

"Our research shows that even the most gifted performers need a minimum of ten years (or 10,000 hours) of intense training before they win international competitions" (Ericsson et al., 2007).

---

## Round 0.2 · accepted · Accept

Thank you for your timely revisions. I've taken time to review your responses, and feel you adequately addressed all concerns. The new analysis is quite interesting as well, and although there are always issues related to interpreting the effect of possible confounds in quasi-experimental work like this, I think that those arguments are left for proponents of the DPM to put forth.

To be fair, I still think there are a number of angles which a critic could attack the current work. The historical graphs show that runners improve for about 5 years after getting competitive, before injury or age leads to a decline. Even though this time as a percentage is relatively small, it is probably true of many fields of expertise that the line between really good and expert is likewise small, and this is sort of the essence of the power law of practice. Also, the fact that this is a highly physical skill may lead proponents to discount it, but at least it places a boundary condition on the DPM theory and suggests that it might actually be falsifiable, which is a substantial critique I have against the DPM; it seems like for any expert, one can always concoct a story about why the DPM applies. Third, I don't think the DPM model is significantly harmed if the estimate of 10 years/10,000 hours is reduced; maybe to 5 years/5,000 hours for this domain; a proponent may simple look at your data as evidence that some 'simpler' domains require less practice to get there, but that the basic theory still applies. But the mission of PeerJ is to get the data out there, and the data makes a compelling argument that needs to be addressed, so in this spirit, I am happy to accept the ms.